# Rapid Adaptation and Remote Delivery of Undergraduate Research Training during the COVID-19 Pandemic

Joanna Yang Yowler [1,†], Kit Knier [2,3,†], Zachary WareJoncas [4,†], Shawna L. Ehlers [2,5], Stephen C. Ekker [2,4], Fabiola Guasp Reyes [6], Bruce F. Horazdovsky [2,4], Glenda Mueller [2], Adriana Morales Gomez [2], Amit Sood [7], Caroline R. Sussman [2,8], Linda M. Scholl [2,9,*], Karen M. Weavers [2,9,*] and Chris Pierret [2,4,*]

1. Department of Research, Mayo Clinic, Jacksonville, FL 32224, USA; yang.joanna@mayo.edu
2. Mayo Clinic Graduate School of Biomedical Sciences, Mayo Clinic, Rochester, MN 55905, USA; knier.catherine@mayo.edu (K.K.); ehlers.shawna@mayo.edu (S.L.E.); ekker.stephen@mayo.edu (S.C.E.); horazdovsky.bruce@mayo.edu (B.F.H.); mueller.glenda@mayo.edu (G.M.); moralesgomez.adriana@mayo.edu (A.M.G.); sussman.carli@mayo.edu (C.R.S.)
3. Mayo Clinic Medical Scientist Training Program, Mayo Clinic, Rochester, MN 55905, USA
4. Department of Biochemistry and Molecular Biology, Mayo Clinic, Rochester, MN 55905, USA; warejoncas.zachary@mayo.edu
5. Department of Psychiatry and Psychology, Mayo Clinic, Rochester, MN 55905, USA
6. Medical Sciences Campus, University of Puerto Rico School of Medicine, San Juan, PR 00936, USA; fabiola.guasp@upr.edu
7. Global Center for Resiliency and Wellbeing, Rochester, MN 55905, USA; as@resilientoption.com
8. Department of Nephrology and Hypertension, Mayo Clinic, Rochester, MN 55905, USA
9. Office of Applied Scholarship and Education Science, Mayo Clinic College of Medicine, Mayo Clinic, Rochester, MN 55905, USA
* Correspondence: Scholl.linda@mayo.edu (L.M.S.); weavers.karen@mayo.edu (K.M.W.); pierret.christopher@mayo.edu (C.P.)
† contributed equally.

**Abstract:** When COVID-19 caused worldwide cancellations of summer research immersion programs in 2020, Mayo Clinic rallied to create an alternate virtual experience called Summer Foundations in Research (SFIR). SFIR was designed not only to ensure the continuance of science pathways training for undergraduate scientists but also to support undergraduate mental wellbeing, given the known pandemic stressors. A total of 170 participants took part in the program and were surveyed pre-post for outcomes in biomedical research career knowledge, biomedical research career interest, research skills confidence, and three dimensions of mental wellbeing. Knowledge of and interest in careers involving biomedical research rose significantly following participation in SFIR. The participants' mean research skills confidence also rose between 0.08 and 1.32 points on a 7-point scale across 12 items from the Clinical Research Appraisal Inventory. Success in science pathways support was accompanied by positive shifts in participant mental wellbeing. Measurable decreases in stress (Perceived Stress Scale, $p < 0.0001$) accompanied gains in resilience (Brief Resilience Scale, $p < 0.0001$) and life satisfaction (Satisfaction with Life Scale, $p = 0.0005$). Collectively, the data suggest that core objectives of traditional in-person summer research programming can be accomplished virtually and that these programs can simultaneously impact student wellbeing. This theoretical framework is particularly salient during COVID-19, but the increased accessibility of virtual programs such as SFIR can continue to bolster science education pathways long after the pandemic is gone.

**Keywords:** undergraduate education; resilience; pandemic; COVID-19

## 1. Introduction

The emergence and spread of COVID-19 continue to alter daily life around the globe. Education is particularly affected by shifts to distance learning and the retooling of school campuses. This change has poignant effects on all aspects of academic life, including the

consequence of increased mental stress reported both in the general population [1–7] and specifically for students [8–12]. The effects on college students have been widely covered in news media [13–15]. The Healthy Minds Study detailed increasing financial stress among college students, as well as worsening depression, academic impairment attributable to mental health, and rising concerns about the future [16].

COVID-19 cancellations of many summer fellowships and internships for undergraduates across the country increased students' uncertainty about their educational opportunities and careers. These programs serve as critical experiential learning tools for many professional development pathways [17,18]. Science, technology, engineering, and mathematics (STEM) summer research fellowships vary by institution, but each plays a key role in student exposure to STEM fields, fosters future opportunities for student professional growth, and boosts recruitment of students to higher learning programs such as medical and graduate school. Coupled with a wider education system attempting to adapt to the pandemic, the loss of summer programs heightens the existing vulnerability of pathways to STEM careers and demonstrates the need for innovative programming to ensure the continuity of postgraduate STEM training.

For the past 30 years, Mayo Clinic has offered 10-week summer undergraduate research programs for students interested in biomedical research training. Typically, students (a) participate in mentored laboratory research and career development workshops; (b) network with peers, laboratory personnel, and faculty; and (c) develop research communication skills. When the pandemic necessitated the elimination of on-campus programming, the graduate school developed a new program for remote delivery. This 4-week experience, Summer Foundations in Research (SFIR), provided the same core academic pillars of a hands-on fellowship while also addressing documented mental health concerns in the participating undergraduate population [11]. The program included education outcomes evaluation and an embedded clinical study to evaluate the achievement of these goals.

SFIR posed the following research questions: (1) Can SFIR programming positively shift participants' attitudes regarding research careers and skills? (2) Can SFIR simultaneously impact learner wellbeing with an embedded stress management intervention? These questions are driven by the underlying belief that high-quality biomedical training can be successfully delivered via distance learning.

## 2. Theoretical Framework

The program objectives targeted four education aims: (1) support the academic trajectory gap in research science created by COVID-19; (2) build sustainable scientific relationships with mentors, peers, and the community; (3) create opportunities for participants to share and address concerns with their own experiences in the pandemic; and (4) provide support for individual wellbeing, given widespread student mental health concerns both preceding and in relation to COVID-19 [11].

These aims intentionally integrate academic achievement with wellbeing support to form the theoretical framework guiding SFIR's design and implementation. COVID-19 not only resulted in the rescindment of summer academic offerings; it also drastically changed the fabric of social life. For undergraduate students, many activities traditionally associated with the collegiate experience were postponed or cancelled. Yet, community and relationship building are key to student academic and life success [12,19]. As such, SFIR made explicit inclusion of opportunities for participants to interact with each other and with mentors, even in a virtual space. Key to facilitation of wellbeing was a focus upon building resilience, defined as the ability to adapt or bounce back in the face of stress or trauma [20]. Although resilience is distinct from wellbeing, it is thought to moderate the relationship between stressors and wellbeing and to predict depression and subjective wellbeing [21]. Building resilience is integral to SFIR's theoretical framework, as this training helps individuals thrive in stressful environments, which is important for student success during and beyond the COVID-19 pandemic.

## 3. Methods

Curriculum development and adaptation for remote delivery: A team of education leaders created the SFIR curriculum by adaptation of existing components and de novo curricular design. Best practice for rapid online transitioning was consulted during SFIR's creation, utilizing many techniques now published as educational recommendations from around the globe [7,19,22,23]. The resulting curriculum consisted of four components: introduction to experimental design, dialogue methodology for communicating science, scientific mentoring, and Stress-Management and Resiliency Training (SMART) [24,25]. This curriculum was adapted for remote delivery via a longstanding Mayo Clinic collaboration with the Integrated Science Education Outreach (InSciEd Out) Foundation [26,27]. The education team then delivered SFIR through a combination of synchronous interactions (scientific presentations, small group discussions, and one-on-one mentoring) and self-paced asynchronous online modules. The syllabus for the program is included in Supplementary Material S1.

The capstone product of the SFIR experience was a presentation at a virtual poster session. The participants summarized their work with lab immersion mentors and their choice of other personally impactful elements from the four-week program. The presenters delivered their poster talks asynchronously using a five-minute screen capture recording, with information uploaded to a shared video database (flipgrid.com/mcsfir2020 accessed on 27 May 2021). The SFIR directors sent invitations to attend the poster session broadly at the institutional level and encouraged participants to directly invite families, friends, and mentors. Poster viewers could also connect directly to Q&A rooms hosted by the presenters for synchronous discussion.

Study design and participant selection: The work presented here is a nonrandomized pre-post case study of the inaugural 2020 SFIR cohort with an external control of undergraduate students for wellbeing measures (see Discussion and Supplementary Material S1). This design was chosen due to sampling and timeframe constraints surrounding the rapid adaptation of Mayo's undergraduate training. Randomization was not completed to ensure equity of access for study participants, who were first and foremost undergraduate biomedical trainees. The participants consisted of a self-selected subset of students previously accepted for in-person summer undergraduate research programs across Mayo Clinic's three campuses. The original selection criteria for in-person programming required students to have completed at least one year at a US college or university, to have a grade point average of at least 3.0 on a 4.0 scale, and to have demonstrated interest in biomedical research. The application deadline was 1 March 2020, and selection was based on academic experience, research experience, a personal statement, and letters of recommendation. Of the 270 eligible students from the initially selected pool, 170 students opted to participate.

Outcomes evaluation: The evaluation team for SFIR gathered critical quantitative and qualitative feedback from participants about the quality and value of each of the program's components. Key educational outcomes tracked in program evaluation included pre-post changes in career understanding, career interest, and confidence in the development of research skills. Evaluation of career metrics used de novo questions, while research skills confidence assessment deployed an adapted subset of 12 items from the Clinical Research Appraisal Inventory [28]. Further details on the survey methodology and analysis are available in Supplementary Material S1.

The embedded clinical study of wellbeing utilized three questionnaires administered pre-post programming to assess the effects on mental resilience (Brief Resilience Scale [29]), stress (Perceived Stress Scale [30]), and life satisfaction (Satisfaction with Life Scale [31]). These questionnaires and the domains they measure were selected due to their validated psychometric properties [29–31], established relevance to SMART training [24,32,33], and their associations with academic, career, and personal success [34,35]. An expanded methods section detailing the number of items, validity testing, scoring, cut-off scores, and statistical analyses can be found in the Supplementary Material S1.

## 4. Results

Educational Outcomes: The participants' knowledge of and interest in careers involving biomedical research rose significantly following SFIR. The proportion of participants indicating they were "very" or "extremely" knowledgeable about such careers jumped from 16% at baseline to 61% at program end. The inclusion of participants "moderately" knowledgeable of careers in biomedical research pushed this statistic to 99% post-SFIR. Parallel to this trend, the proportion of participants indicating high levels of interest in pursuing biomedical research careers (5 or 6, on a 0 to 6 scale) increased substantially over the course of the program, starting at 33% and ending at 73%. Finally, at program end, 85% of participants indicated they were considering applying to a Mayo Clinic education program; only 4% responded that they were not considering applying, with the remaining 11% reporting uncertain application plans.

Most notable were the gains participants made in confidence across the 12 key research skills measured. Across all skills, the participants' mean confidence levels rose between 0.08 to 1.32 points on a 7-point scale (Figure 1). The strongest gains in mean confidence levels were seen in designing a study and collaborating with others.

The post-program survey of the participants gathered quantitative and qualitative feedback about program strengths and areas for improvement. The program components that were particularly highly rated included the SMART sessions and mentoring. SMART sessions were rated "quite" or "extremely" worthwhile by 85% of the participants. Respectively, 99% percent and 96% of the students indicated that their mentors were supportive and showed genuine interest in their research ideas.

Wellbeing Outcomes: SFIR participants demonstrated gains across all three dimensions of wellbeing. The responses on the Brief Resilience Scale indicate improved resilience after program participation ($M$ ($SD$) pre- 3.30 (0.68), post- 3.51 (0.68), $\Delta$ + 0.21 (0.55), $t(128) = 4.41$, $p < 0.0001$). At the same time, these learners reported decreases in stress on the Perceived Stress Scale ($M$ ($SD$) pre- 19.98 (6.89), post- 18.06 (6.33), $\Delta$ −1.91 (5.27), $t(124) = -4.06$, $p < 0.0001$). They also recorded increases in life satisfaction, as measured by the Satisfaction with Life Scale ($M$ ($SD$) pre- 24.10 (6.03), post- 25.43 (6.31), $\Delta$ +1.33 (4.29), $t(130) = 3.54$, $p = 0.0005$). These results correspond to small Cohen's $d$ effect sizes [36] in all dimensions (resilience $d = 0.38$; stress $d = 0.36$; life satisfaction $d = 0.31$). Wellbeing trends utilizing established inventory cut-offs are visualized in Figure 2. There is a desirable shift toward normal to high resilience, low to moderate stress, and general to extreme satisfaction with life.

A qualitative assessment of participant feedback found the overwhelmingly positive reception of SMART—the programmatic component most explicitly tied to wellbeing. In addition to being a highlight of the program for many participants, SMART's mindfulness training was interpreted to be important for personal and professional development. Representative comments include:

"Mindfulness was incredibly useful because of how it gave me a different perspective on how to address stress and issues in my life"

and

"Dr. [X's] mindfulness sessions were a highlight of the program for me. He gave really concrete and valuable advice for improving relationship(s) and [how to] have a positive, well adjusted mindset. All of these are highly valuable in a scientific career."

These comments reinforce the results from the wellbeing surveys and indicate the effectiveness of integrating stress-management training into STEM education programming.

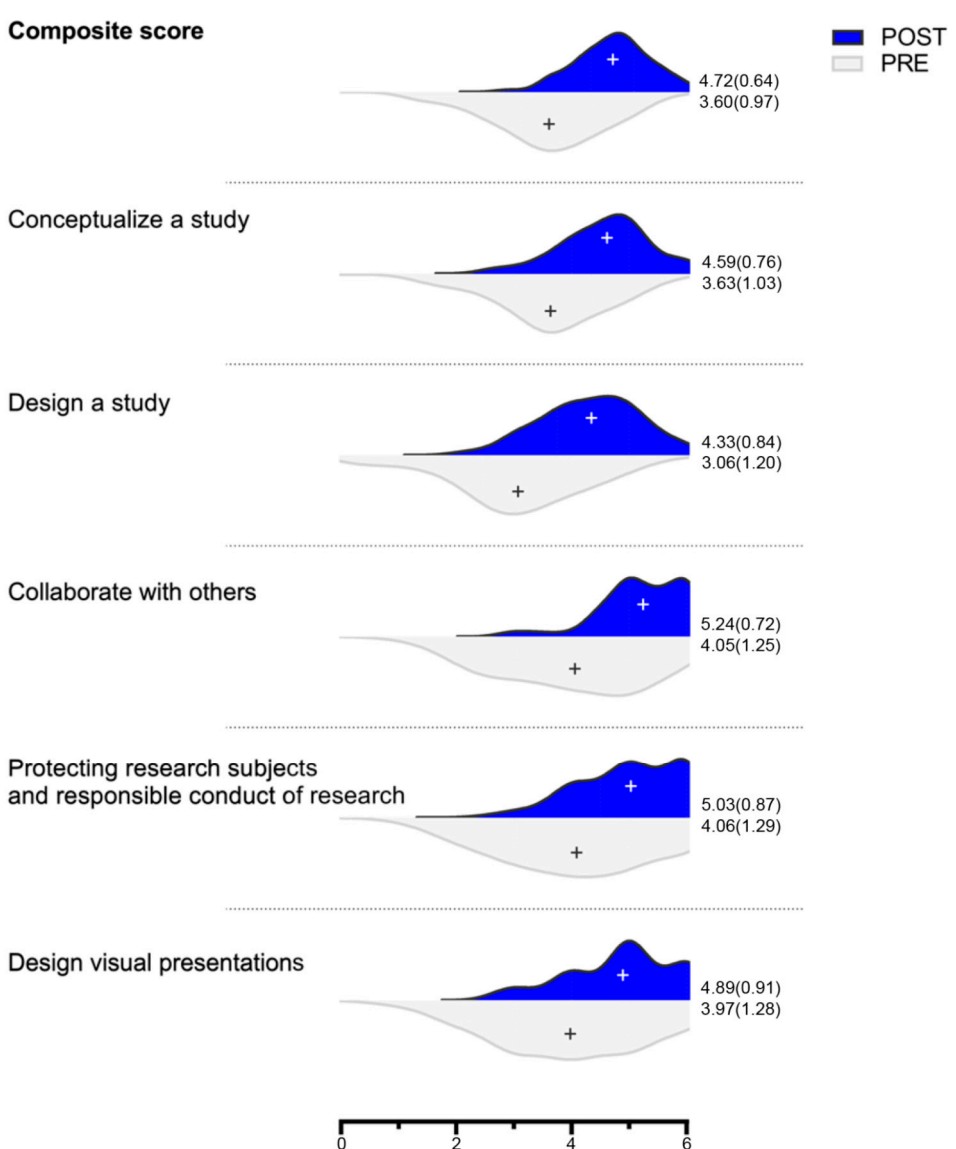

**Figure 1.** Twelve items belonging to one of five categories were selected from the Clinical Research Appraisal Inventory (CRAI), prior to the beginning of the Summer Foundations in Research (SFIR) program. The items were rated on a scale of 0 to 6, from no confidence at all to total confidence. Mean is indicated as "+" and $M$ ($SD$) are given to the right of each distribution. The composite score across the 12 items had a $M_{dif}$ ($SD_{dif}$) of 1.11 (0.79) ($n = 141$). Conceptualize a study (select a suitable topic area, articulate a clear purpose for the research, refine a problem so it can be investigated) 0.97 (0.94) ($n = 144$); design a study (compare major types of studies, choosing an appropriate design to test hypotheses, select appropriate methods of data collection, design the best data analysis strategy) 1.26 (1.17) ($n = 144$); collaborate with others (consult a senior researcher for ideas, participate in generating collaborative research) 1.20 (1.07) ($n = 143$); protect research subjects and responsible conduct of research (discuss ethical issues in research conduct, identify institutional responsibilities in research conduct) 0.97 (1.05) ($n = 145$); and design visual presentations 0.92 (1.12) ($n = 145$).

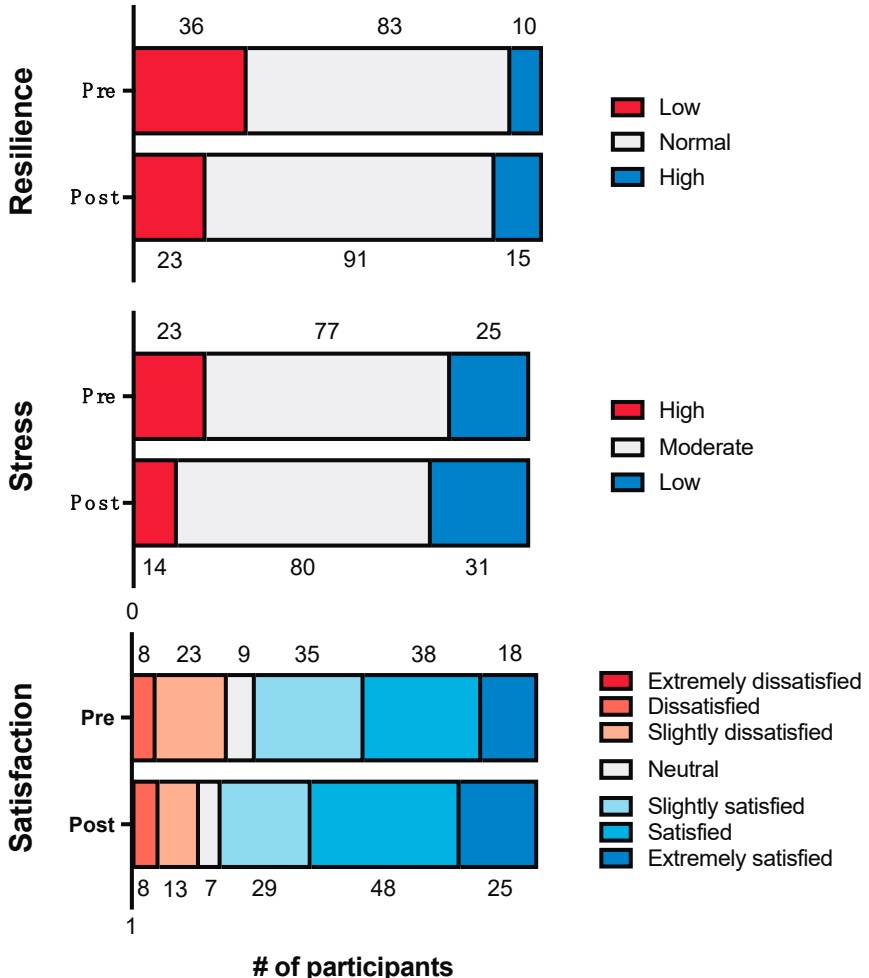

**Figure 2.** Participant wellbeing responses pre-post Summer Foundations in Research (SFIR) programming were plotted for distribution analyses. Score cut-offs for categorization as follows: Brief Resilience Scale *n* = 129 | Low (1.00–2.99), Normal (3.00–4.30), High (4.31–5.00); Perceived Stress Scale *n* = 125 | High (27–40), Moderate (14–26), Low (0–13); Satisfaction with Life Scale *n* = 131 | Extremely Dissatisfied (5–9), Dissatisfied (10–14), Slightly Dissatisfied (15–19), Neutral (20), Slightly Satisfied (21–25), Satisfied (26–30), Extremely Satisfied (31–35). SFIR students report gains across all three wellbeing categories pre-post programming.

## 5. Discussion

Positive educational outcomes for SFIR participants in measurements of career understanding, career interest, and research skills confidence reveal that many goals of research training can be meaningfully addressed in a digital setting. Prior to the delivery of SFIR, Mayo Clinic faculty voiced concerns that the value of summer undergraduate research would be lost without in-person interactions. Major concerns included how participants would develop laboratory skills without setting foot in a laboratory and how mentors would be able to form meaningful connections with young scientists without face-to-face apprenticeships. The results presented here provide evidence that such fears can be allayed with intentionally designed programming. At the completion of SFIR, the participants showed confidence growth in core research skills across all of the measured domains of study conceptualization, study design, research collaboration, responsible conduct of research, and data presentation. Mentor/mentee relationships flourished in a digital setting. The participants were actively engaged in small peer groups that helped them feel connected to each other and to the program faculty and facilitators. These factors likely contributed to the observed increases in career knowledge and interest.

Improvements in wellbeing metrics of resilience, stress, and satisfaction with life accompanied the above educational gains. This is noteworthy because equipping the next generation of medical and graduate students with tools to decrease stress and improve resilience—two major features of burnout—is of great interest to the academic community [37–39]. Moreover, a greater satisfaction with life among students is paramount to education as a whole and is a cornerstone for cultivating scientific excellence in a wellness environment [40]. Although the effect sizes of wellbeing gains were modest, they are similar to previous interventions that deployed the SMART program [24,34]. It is important to note that SMART was a mandatory component of SFIR, which has been shown to restrict effect sizes when compared with opt-in studies [32].

To increase confidence in the observed wellbeing results, given the turbulent social and political climate of 2020, an external control group of previous Mayo Clinic summer undergraduate students was recruited. The selected students were current sophomores or juniors in the 2020–21 school year and did not receive SMART during their time at Mayo—making them an ideal control for the 2020 SFIR cohort. In comparison to this external control (Supplementary Material S1), SFIR students had statistically significant (Mann–Whitney U test) improvements in resilience ($p = 0.03$) and decreases in stress ($p = 0.03$). There was not a significant difference in the gains observed for satisfaction with life ($p = 0.81$). These results strengthen the assertion that SFIR programming positively impacted student wellbeing even in light of the tumultuous pandemic.

## 6. Limitations

The results from this study are limited by its participant selection criteria and short-term analysis. Regarding participant selection, SFIR students included a diverse representation (see Supplementary Material S1) of undergraduates pursuing clinical and translational biomedical research—despite being a selected cohort. In the light of demographic breadth of SFIR participants, the study results are likely generalizable to undergraduates engaged in research through US programs. Of interest to education communities at every level is exactly those who might struggle and/or benefit the most in a distance-learning environment. Future analyses will consider sociodemographic subgroups of students to address the known educational and health disparities. Limitations regarding short-term analysis can be addressed by future longitudinal follow-up. Preliminary data at three months post-SFIR showed positive outcomes, and future analyses of sustained impact will help elucidate whether there is a need for maintenance efforts following programs like SFIR. Such data will also provide key insights into equity and inclusion in distance learning for undergraduate students.

## 7. Conclusions, Implications, and Future Directions

Due to the benefits shown in this digital format, SFIR will see continual implementation at Mayo Clinic. The program is being adapted to act as an onboarding experience prior to undergraduate researchers' arrival for face-to-face mentorship and as a stand-alone offering to increase the reach of Mayo Clinic's science programming. An abbreviated version preceding summer or school year research aims to enhance the confidence and preparedness of undergraduates for their first experience at a research-focused institution. Furthermore, the program will provide a peer group for support across laboratories, while simultaneously enhancing opportunities for near-peer mentoring by graduate students and postdoctoral fellows.

COVID-19 has challenged learners and educators across the world with its accelerated demand for digital learning. The path of least resistance over the pandemic year has often resulted in the cancellation of important educational and professional development programs. While such measures are sometimes unavoidable, a more sustainable approach is to treat COVID-19 as an opportunity for growth—preparing for inevitable future disruptors of the education system.

The results from the SFIR case study show that many of the goals of in-person undergraduate biomedical sciences training can be achieved (or even exceeded) in a virtual setting. These goals go beyond the strictly didactic, as virtual programming can simultaneously target and improve the student wellbeing essential to academic, career, and life satisfaction and success. Such a holistic support has received increased focus given the COVID-19 pandemic's documented consequences for mental health; however, the strategies employed here will continue to improve undergraduate life long after COVID-19 is gone. In the end, SFIR is one example of how STEM disciplines can embrace adaptation, preserving the integrity of pathways to science, and in doing so, celebrating the spirit of scientific innovation.

**Supplementary Materials:** The following are available online at https://www.mdpi.com/article/10.3390/su13116133/s1, SFIR Syllabus, Unabridged Methods, Table S1: External Control Summary, Table S2: Demographic Summary of Enrolled Participants, SFIR Dialogue Templates, SFIR 1:1 Mentorship Expectations, SFIR Qualitative Analysis Examples (Mindfulness), and SFIR Datasets.

**Author Contributions:** Conceptualization, J.Y.Y., K.K., Z.W., A.S., L.M.S., K.M.W. and C.P.; methodology, J.Y.Y., K.K., S.C.E., L.M.S., K.M.W. and C.P.; formal analysis, J.Y.Y., K.K., S.L.E., F.G.R., A.M.G. and C.P.; investigation, J.Y.Y., K.K., A.S., C.R.S., L.M.S. and C.P.; resources, B.F.H. and G.M.; data curation, L.M.S.; writing–original draft, J.Y.Y., K.K., Z.W., F.G.R., A.M.G., A.S., C.R.S., L.M.S., K.M.W. and C.P.; writing–review & editing, J.Y.Y., K.K., Z.W., S.L.E., S.C.E., F.G.R., B.F.H., G.M., A.M.G., A.S., C.R.S., L.M.S., K.M.W. and C.P.; visualization, K.K.; supervision, J.Y.Y., S.C.E., B.F.H., L.M.S. and C.P.; project administration, Z.W., B.F.H., G.M., K.M.W. and C.P.; funding acquisition, Z.W., G.M. and C.P. All authors have read and agreed to the published version of the manuscript.

**Funding:** This research was funded by the William Randolph Hearst Foundation, other benefactors, and CTSA Grant Number UL1 TR002377 from the National Center for Advancing Translational Sciences (NCATS), a component of the National Institutes of Health (NIH).

**Institutional Review Board Statement:** The study was conducted according to the guidelines of the Declaration of Helsinki, and approved by the Institutional Review Board of Mayo Clinic (ID: 20-00657, 20 July 2020). It was declared exempt under 45 CFR 46.104d, Category 2.

**Informed Consent Statement:** Informed consent was obtained from all subjects involved in the study.

**Data Availability Statement:** The data presented in this study are available in Supplementary Material S2 and S3 (academic and clinical data, respectively).

**Acknowledgments:** The SFIR program was made possible by the William Randolph Hearst Foundation and other benefactors whose generous support of undergraduate research opportunities helps create tomorrow's physicians and scientists. SFIR relied on the partnership and expertise of the Integrated Science Education Outreach (InSciEd Out) Foundation to convert curriculum for digital delivery. This work was made possible by CTSA Grant Number UL1 TR002377 from the National Center for Advancing Translational Sciences (NCATS), a component of the National Institutes of Health (NIH). Its contents are solely the responsibility of the authors and do not necessarily represent the official view of NIH.

**Conflicts of Interest:** The authors declare no conflict of interest.

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
