# Peer review of "Rapid Adaptation and Remote Delivery of Undergraduate Research Training during the COVID-19 Pandemic"

_sustainability, doi:10.3390/su13116133_

Round 1

Reviewer 1 Report

I am glad to review this informative study, which presents a new idea entitled, “Rapid adaptation and remote delivery of undergraduate research training during the COVID 19 pandemic.” It is a very new and vital research direction. The authors have presented a novel issue in this article. The all-purpose writing of this manuscript is clear, and the idea of the article is creative. This article aims to explore an impressive research topic of the COVID-19 impacts on learning process and undergraduate research training.

1. I suggest authors avoid passive style in the Abstract, and it must be high quality as it the "FACE" of the study.
2. Add few lines in the introduction and literature sections on how social media and internet use among students is helpful.
3. Add few lines on the contribution of this study and how the results are insightful for academic purposes.
4. Add separate heading and discuss implications . Add separate heading and discuss limitations 
5. Pay attention to English quality to reach scientific merit.

Suggested studies:
Maqsood, A., Abbas, J., Rehman, G., & Mubeen, R. (2021, 2021/11/01/). The paradigm shift for educational system continuance in the advent of COVID-19 pandemic: Mental health challenges and reflections. Current Research in Behavioral Sciences, 2, 100011. https://doi.org/10.1016/j.crbeha.2020.100011
Abbas, J., Aman, J., Nurunnabi, M., & Bano, S. (2019). The Impact of Social Media Learning Behavior for Sustainable Education: Evidence of Students from Selected Universities in Pakistan. Sustainability, 11(6), 1683.  

Author Response

Thank you so much for your prompt and helpful review. I have attached our response to all reviewers so you may see the overall changes including those you have requested.

Reviewer 2 Report

The paper provides an interesting description of the Mayo Clinic’s experience, showing how it has been able to react to Covid-19 crisis by shifting to distance learning and organizing a research training for undergraduate students.

Findings show the program success in terms of positive educational and wellbeing outcomes for participants.

Despite the relevance of such topic, the aims pursued through the study are not very clear: what is your aim? Which are your research questions? And what are the practical contributions/implications related to your study?

There is no section dedicated to methodology: you performed a case study analysis, why? Can you describe your data collection and data analysis processes? What are the main threats and weaknesses characterizing such method?

There is no theoretical framework in which your analysis can be inserted providing a contribution; in my opinion, this work should be based on an analysis of distance learning from a literature point of view.

Furthermore, you considered the word “resilience” as one of the keywords… Why? And what is resilience? Why distance learning and the SFIR experience can be presented as resilience strategies?

I suggest dedicating a theoretical section to the concept of resilience.

For what concerns conclusions, I suggest, again, to better specify what are the main implications linked to the study.

Furthermore, it would be interesting to add considerations in terms of future perspectives: could distance learning become an ordinary practice for Mayo Clinic? Will the SFIR be replicated? Or is it destined to remain a single experience?

Best wishes

Author Response

(The authors gave the same response as above.)

Reviewer 3 Report

Nowadays, more and more studies analyzing the impact of COVID-19 on the educational system are conducted worldwide. I believe, every research helps to see a broader picture and to adapt to these new conditions.

Thus, I find the paper interesting and useful. The main research question concerning the educational process organization in an electronic environment is of vital importance nowadays considering the poignant effects of Covid-19. What I personally found interesting is the fact that besides the online educational process organization, the authors provided an opportunity for Stress-Management and Resiliency Training.

The paper is clear and easy to read. The Supplementary Materials provided give a better understanding of the research conducted by the authors.

Some minor corrections may be done. The structure of the paper may be changed a little bit. I recommend adding "Materials and Methods", "Limitations", and "Conclusion" sections. It can be easily done, as all the necessary information is already given in the paper.

Author Response

(The authors gave the same response as above.)

Round 2

Reviewer 2 Report

The manuscript has significantly improved with respect to the previous version and can now be accepted in its current form. Congratulations!